# Polyzwitterionic Coating of Porous Adsorbents for Therapeutic Apheresis

**DOI:** 10.3390/jfb13040216

**Published:** 2022-11-03

**Authors:** Vladislav Semak, Tanja Eichhorn, René Weiss, Viktoria Weber

**Affiliations:** Department for Biomedical Research, Center for Biomedical Technology, University for Continuing Education Krems, 3500 Krems, Austria

**Keywords:** polyzwitterions, coat, adsorbents, extracorporeal therapies, blood compatibility

## Abstract

Adsorbents for whole blood apheresis need to be highly blood compatible to minimize the activation of blood cells on the biomaterial surface. Here, we developed blood-compatible matrices by surface modification with polyzwitterionic polysulfobetainic and polycarboxybetainic coatings. Photoreactive zwitterionic terpolymers were synthesized by free-radical polymerization of zwitterionic, photoreactive, and fluorescent monomers. Upon UV irradiation, the terpolymers were photodeposited and mutually crosslinked on the surface of hydrophobic polystyrene-*co*-divinylbenzene and hydrophilic polyacrylamide-*co*-polyacrylate (DALI) beads. Fluorescent microscopy revealed coatings with an average thickness of 5 µm, which were limited to the bead surface. Blood compatibility was assessed based on polymer-induced hemolysis, coagulation parameters, and in vitro tests. The maintenance of the adsorption capacity after coating was studied in human whole blood with cytokines for polystyrene beads (remained capacity 25–67%) and with low-density lipoprotein (remained capacity 80%) for polyacrylate beads. Coating enhanced the blood compatibility of hydrophobic, but not of hydrophilic adsorbents. The most prominent effect was observed on coagulation parameters (e.g., PT, aPTT, TT, and protein C) and neutrophil count. Polycarboxybetaine with a charge spacer of five carbons was the most promising polyzwitterion for the coating of adsorbents for whole blood apheresis.

## 1. Introduction

Adsorbent-based extracorporeal therapies are clinically established to deplete pathogenic factors, such as lipoproteins, toxic metabolites, protein-bound substances, as well as autoantibodies from the circulation of patients suffering from severe hypercholesterolemia, liver dysfunction, or autoimmune diseases [1,2,3]. Beyond that, extracorporeal approaches are emerging for the depletion of inflammatory mediators in patients suffering from systemic inflammation and hypercytokinemia (cytokine storm) [4,5,6]. Independent of the targeted molecules and the adsorption mechanism, adsorbents immediately acquire a surface layer of plasma proteins, such as albumin and immunoglobulins, as well as complement and coagulation factors upon contact with whole blood. This protein layer forms a new and dynamic interface between the biomaterial and the blood, a phenomenon known as the Vroman effect [7,8]. Immune cells, platelets, and red blood cells sense the biomaterial surface through this protein layer. Depending on the characteristics of the biomaterial, adsorbed proteins may undergo conformational changes, which can activate the contact, coagulation, and complement systems and trigger immunological reactions with potentially detrimental consequences for the host and the biomaterial function.

Blood compatibility is particularly important in therapeutic hemapheresis, where adsorbent polymers are in direct contact with whole blood. We have previously characterized the relationship between the physicochemical characteristics of adsorbent polymers used for hemapheresis and cellular activation at the blood–biomaterial interface [9,10,11,12,13]. Here, we aimed to develop blood-compatible zwitterionic polymers that can be applied as surface coatings to enhance the blood compatibility of porous adsorbents.

Zwitterions are formally neutral compounds with single opposite charges on nonadjacent atoms. Polyzwitterions are polymers containing these opposite ionic groups on the same monomeric unit [14,15,16,17]. Their application for non-thrombogenic surface modifications is inspired by the external surface of mammalian cell membranes composed of phosphorylcholine residues [18], and the reduction in protein adsorption on zwitterionic surfaces is mainly attributed to the lack of surface-associated ions [19,20,21,22,23]. These characteristics of polyzwitterions, also known as polybetaines, give rise to a broad spectrum of potential biomedical applications [24,25]. However, till today, only a few attempts have been performed in the field of surface adsorbent modifications by polyzwitterions and these efforts are mainly focused on absorbents for wastewater treatment [26,27]. Here, we assessed the effect of sulfobetaine (SBE) and carboxybetaine (CBMA) polyzwitterionic coatings (Figure 1) on the blood compatibility of a hydrophobic polystyrene-based polymer as well as on a hydrophilic polyacrylate-based polymer.

Both adsorbents consist of sponge-like porous beads with pores offering a large inner surface for the adsorption of their target molecules, while this inner surface is not accessible for blood cells. Since blood cells and platelets are in contact only with the outer surface of the adsorbent beads, we aimed to limit the coating to the external adsorbent surface to yield matrices with enhanced blood compatibility, while maintaining their adsorption capacity and selectivity. The selection of polyzwitterions was made with the aim to cover different behaviors of polyzwitterions, such as head-group hydrophilicity (carboxylate is more hydrophilic than sulfonate) or pKa values. In the case of carboxybetaines, pKa is dependent on the number of methylene groups separating the charged sites [28]. We employed UV light-induced photodeposition of zwitterionic copolymers containing a photolabile aryl azide moiety [29,30,31,32] as a grafting-to method. This approach does not require prior functionalization of the surface or additional reagents, thereby avoiding any contamination of the adsorbents. To monitor the deposition and stability of the coating on the bead surface, we incorporated a green fluorescent nitrobenzoxadiazole (NBD) [33] label. The adsorption capacity and the blood compatibility of the resulting coated polymers were characterized in vitro using human whole blood. We hypothesized that photodeposited polyzwitterionic coatings would enhance the blood compatibility of adsorbents without drastically decreasing their adsorption capacity since the coats are limited to the outer surface.

## 2. Materials and Methods

A list of all chemicals (reagents, buffers, priming solutions), information on venipuncture and anticoagulation of human whole blood, description of flow cytometry, fluorescent and confocal laser scanning microscopy, scanning electron microscopy (SEM), energy-dispersive X-ray spectroscopy (EDX), Fourier-transform infrared spectroscopy (FTIR), and statistical analysis as well as the synthesis of monomers and copolymers are given in Appendix A.

### 2.1. Abbreviations

The following names and abbreviations of monomers and copolymers will be used in the whole text. *Monomers*: CBMA-C1, 2-[dimethyl-[3-(2-methylprop-2-enoylamino)propyl]azaniumyl]acetate; CBMA-C5, 6-[dimethyl-[3-(2-methylprop-2-enoylamino)propyl]azaniumyl]hexanoate; SBE = *N*-(3-sulfopropyl)-*N*-methacroyloxyethyl-*N*,*N*-dimethylammonium betaine; Az-MAAm = 4-azidophenyl methacrylamide; NBD-MAA, nitrobenzoxadiazoleaminoethyl methacrylate. *Copolymers*: pSBE = poly[SBE-*co*-Az-*co*-NBD], pCBMA-C1 = poly[CBMA-C1-*co*-Az-*co*-NBD]; pCBMA-C5, poly[CBMA-C5-*co*-Az-*co*-NBD]. For the sake of brevity, coated adsorbents are labeled as adsorbent-copolymer, e.g., “CG161c-pSBE” means “Amberchrom CG161c beads coated with poly[SBE-*co*-Az-*co*-NBD] terpolymer”.

### 2.2. Adsorbents

DALI (“Direct Adsorption of Lipoproteins”, Fresenius Medical Care, Bad Homburg, Germany) consists of polyacrylamide beads (particle size 150–200 µm, surface area 50 m^2^/g, mean pore diameter 180 nm) functionalized with poly(acrylic acid). It binds to lipoproteins containing apolipoprotein B100 (apoB100) moieties, including low-density lipoprotein (LDL) and lipoprotein(a), via electrostatic interactions and is clinically applied to treat patients suffering from homozygous familial hypercholesterolemia [3,34,35,36]. CG161c (Amberchrom CG161c, Dow Chemical, Philadelphia, PA, USA) consists of polystyrene-divinylbenzene beads (average particle size 120 µm, surface area 900 m^2^/g, mean pore diameter 15 nm). CytoSorb (CytoSorbents Corporation, Monmouth Junction, NJ, USA) is a polystyrene-divinylbenzene-based adsorbent (particle size 400–600 µm, surface area 850 m^2^/g, pore size 0.8–5 nm) coated with polyvinylpyrrolidone. It is clinically applied for cytokines depletion in patients suffering from sepsis, COVID-19 [37,38,39,40], or to dampen hypercytokinemia during surgery [41,42,43]. It binds a range of cytokines via hydrophobic interactions; however, it exhibits reduced adsorption of cytokines of higher molecular mass, such as tumor necrosis factor-alpha (TNF-α, 51 kDa) [44,45]. In this study, CytoSorb was employed as a positive control in cytokine depletion experiments.

### 2.3. Pre-conditioning

All adsorbents were washed prior to use (e.g., coating, blood compatibility, and adsorption experiments). CG161c was conditioned by washing with ethanol in saline solution, using a gradient from 96% to 0%. DALI and CytoSorb were extensively washed with saline solution. Washed adsorbents were stored as 50% suspension (vol%) in 0.9% sterile NaCl at 4 °C.

### 2.4. Physical Coating and UV Crosslinking

Identical coating procedures were applied for all types of adsorbents and polyzwitterions (Figure 2). The general coating protocol included (1) the preparation of a bead suspension in an aqueous solution of coating terpolymer (1% wt); (2) physical pre-coating of the beads by solvent evaporation; (3) UV photocrosslinking; and (4) conditioning and washing steps. The coating of CG161c with pSBE is described as follows: A slurry of conditioned beads (20 mL, 4.50 g bead weight) in pSBE copolymer (45 mg, 1% wt.) aqueous ethanol solution (14% vol) was placed onto a glass Petri dish (d = 18.5 cm) and shaken (400–500 rpm) for 48 h at room temperature. After 24 h, attached beads were gently removed from the plate bottom with a plastic spatula. Physical pre-coating was performed in the dark to minimize side reactions or bleaching of the fluorescent label. Pre-coated CG161c-pSBE beads (10 mL, approximately 2.85 g) were transferred to a new glass Petri dish, which was placed in a shaker at 20 cm distance from a UV lamp (254 nm, Osram Puritec HNS 15W G13, OSRAM GmbH, Munich, Germany). After shaking (400–500 rpm) for 20 min at room temperature under UV light (210 lux; background illuminance 35 lux), the crosslinked beads were washed with a gradient ranging from 96% to 0% ethanol in saline solution. Ten washing cycles were performed, and after the 4th to 6th washing step, polyzwitterions were undetectable by UV/Vis spectroscopy (Ultrospec 3300 pro, Amersham Biosciences, Little Chalfont, UK) in the supernatant at λ_max_ = 468 nm.

### 2.5. Characterization of Blood Compatibility

To assess the blood compatibility of the coated adsorbents, we focused on coagulation parameters, including prothrombin time (PT), activated partial thromboplastin time (aPTT), thrombin time (TT), fibrinogen (Fg), antithrombin III (AT III), and Protein C, as well as polymer-induced hemolysis. Uncoated DALI and CG161c beads were used as controls.

Adsorbents (200 µL per tube) were placed into vacutainer tubes (Greiner Bio-One, 2 mL, without additives) in a saline suspension (50% vol). The beads were pelleted (70× *g*, 5 min), and the supernatant was discarded. Plasma (1800 µL) was added (10% vol), and the suspension was incubated for 60 min at 37 °C with gentle rotation; (n = 3). The beads were pelleted at 2000× *g*, 10 min, room temperature, and the plasma was collected. Plasma samples from tests with CG161c-based adsorbents were additionally centrifuged (4000× *g*, 5 min) to avoid possible sample contamination with adsorbent debris. All samples were immediately analyzed for activated partial thromboplastin time (aPTT), prothrombin time (PT), and thrombin time (TT). In addition, aliquots stored at −20 °C were analyzed for antithrombin III (AT-III), fibrinogen, and protein C. All measurements were performed on a coagulation analyzer (Sysmex CA-560, Siemens Healthineers, Erlangen, Germany; n = 3). Details on analytical reagents are given in the Appendix A.

Polymer-induced hemolysis was evaluated in whole blood anticoagulated with sodium citrate. Polyzwitterions up to concentrations of 1.0 mg/mL or coated adsorbents (10% vol) were incubated in whole blood for 4 h at 37 °C, and cell-free hemoglobin (fHb) was determined by the spectrophotometric method of Kahn according to the equation in undiluted plasma [46].
$$fHb\;\left(mg/dL\right)=\left(absorbance_{578}\times 155\right)-\left(absorbance_{562}\times 86\right)-\left(absorbance_{598}\times 69\right)$$

### 2.6. Blood Compatibility of Coated CG161c Beads

Adsorbents (200 µL per tube) were placed into vacutainer tubes (Greiner Bio-One, Kremsmünster, Austria; 2 mL, without additives) in a 50% (vol%) saline suspension. The beads were pelleted (70× *g*, 5 min), and the supernatant was discarded. Whole blood (1800 µL) anticoagulated with sodium citrate was added (10% vol), and the suspension was incubated for 60 min at 37 °C with gentle rotation; (n = 3). The beads were removed by filtration (pluriStrainer, mesh 40 µm, pluriSelect, Leipzig, Germany) and blood count was immediately acquired from the filtrate (blood). Beads were washed with saline, fixed with glutaraldehyde (2.5%), washed with water, frozen, lyophilized, and gold sputtered for SEM evaluation.

### 2.7. Recirculation of Whole Blood over DALI Columns

Human whole blood was anticoagulated with ACD-A 1:20 and 0.8 IU/mL heparin, and aliquots of 50 mL were recirculated over columns containing the DALI adsorbent as shown in Appendix A. For this purpose, adsorbent columns (h = 22 mm, d = 17 mm, adsorbent bed volume 5.0 mL; flow rate 1.2 mL/min) were packed with DALI beads, rinsed with 2 × 20 mL of priming solution containing ACD-A (1:40). Heparin (10 IU/mL) was supplemented during the first rinsing step, as recommended by the manufacturer. Recirculation was performed for 4 h using medical-grade tubing sets and a hemodialysis roller pump. Samples for flow cytometric analysis were drawn from the circuits at the start of each experiment and after 1, 2, and 4 h. Blood cell counts were obtained every 30 min.

### 2.8. Stimulation of Whole Blood with Lipopolysaccharide (LPS)

Human whole blood anticoagulated with sodium heparin was stimulated with 100 ng/mL *E. coli* LPS (O55:B5, purified by gel filtration chromatography, Sigma-Aldrich currently Merck, Darmstadt, Germany) for 4 h at 37 °C [47]. After stimulation, the blood was centrifuged (2000× *g*, 10 min, room temperature) and the resulting plasma was stored at −20 °C until further use.

### 2.9. Cytokine Adsorption and Quantification

Adsorbents (200 µL per tube) were placed into vacutainer tubes (Greiner Bio-One, 2 mL, without additives) in a saline suspension (50% vol). The beads were pelleted (70× *g*, 5 min), and the supernatant was discarded. Plasma (1800 µL) derived from LPS-stimulated blood was added (10% vol), and the suspension was incubated for 60 min at 37 °C with gentle rotation; (n = 3). The beads were pelleted at 2000× *g*, 10 min, room temperature, and the plasma was collected, aliquoted, and kept at −20 °C until further use. Interleukin-6 (IL-6) and tumor necrosis factor-alpha (TNF-α) were quantified by magnetic bead array analysis (Bio-Plex 200, Bio-Rad, Vienna, Austria).

### 2.10. Adsorption of Lipoproteins

Human whole blood was spiked with low-density lipoprotein (LDL from human plasma, Lee Biosolutions, Maryland Heights, MO, USA) to reach an LDL concentration of 200 mg/dL. Aliquots (200 µL) of the DALI beads (preconditioned as described in 2.7) were incubated with the spiked whole blood (1800 µL, 10% vol) at 37 °C for 60 min with gentle rotation; (n = 3). Adsorbents were pelleted by centrifugation (2000× *g*, 10 min, room temperature), and LDL was quantified in the supernatant using the Hitachi Cobas c311 Chemistry Analyzer with the LDL-C reagent set (Roche Diagnostics, Rotkreuz, Switzerland). All experiments were carried out in triplicates.

## 3. Results

### 3.1. Synthesis of Monomers and Copolymers

All polyzwitterions employed in this study are statistical terpolymers possessing zwitterionic (94%), photolabile (5%), and fluorescent (1%) moieties. For the synthesis of polysulfobetaine, commercially available methacrylate SBE was used. Carboxybetaine (CBMA) monomers were synthesized in-house as methacrylamides. The synthesis of these monomers is based on the quaternization of tertiary amines by esters of halocarboxylic acid followed by ester hydrolysis (Appendix A) [48]. Two types of CBMA monomers with one (C1) or five carbon (C5) spacers between positive and negative charges were prepared. The structures of the CBMA monomers were confirmed by FTIR, and their spectroscopic data were in agreement with the literature [48,49].

The key FTIR observation was the band shift in the carboxylate ion antisymmetric C = O stretching, depending on the spacer length, exhibiting a shift from 1616 cm^−1^ in CBMA-C1 to 1570 cm^−1^ in CBMA-C5 (Appendix A) [28]. A photolabile aryl azide monomer (Az-MAAm) was synthesized with high yield by acylation of 4-azidoaniline. FTIR analysis confirmed the formation of amidic bonds (1659 cm^−1^) without affecting the azide functional groups (2112 cm^−1^) (Appendix A) [50]. The presence of amide was confirmed using the colorimetric ninhydrin test [51]. Nitro-2,1,3-benzoxadiazol (nitrobenzofurazan, NBD) fluorescent monomer was prepared in two steps. Due to the electrophilic character (10 π electrons) of its nitro-2,1,3-benzoxadiazole heteroaromatic ring, 4-chloro-7-nitrobenzofurazan easily undergoes nucleophilic aromatic substitution reactions (SN_Ar_) with ethanolamine, and the reaction proceeds within one hour at room temperature. The corresponding methacrylate was prepared by acylation with methacryloyl chloride. The synthesis of all polyzwitterionic copolymers was performed under similar experimental conditions. The feeding ratio of monomers (zwitterionic unit: Az-MAAm:NBD-MA) was 94:5:1, and copolymerization was initiated with α,α′-azobisisobutyronitrile (AIBN, 2%) at 60 °C for 18 h in a methanol–water mixture (Figure 1). Due to the solubility of the employed zwitterionic monomers, the percentage of methanol in water varied from 30% to 50%. Attempts using dimethyl sulfoxide as a co-solvent were not successful. Copolymers were obtained after dialysis (molecular weight cut-off 3.5 kDa) against water in a yield of 40–50%.

The synthesized copolymers were characterized by FTIR and UV/Vis spectroscopy. The disappearance of the alkene (=CH_2_) bond peak at ≈930 cm^−1^ present in the spectra of monomers indicated the formation of the polymer backbone. As noted for carboxybetaine monomers, a shift in the carboxylate band (1621 cm^−1^ pCBMA-C1 vs. 1567 cm^−1^ pCBMA-C5) caused by a different tether group was observed for copolymers, as well (Appendix A). The presence of photolabile aryl azide (Az) moieties was confirmed by the appearance of the characteristic peak for azide (–N_3_) functional groups at ≈2120 cm^−1^. UV/Vis adsorption spectra of all polyzwitterions have adsorption maxima in the 468 nm region. This peak is attributed to the amine-substituted NBD. A direct quantification of aryl azide moieties by FTIR was not appropriate due to the low feeding ratio (5 mol%). To overcome this issue, fluorescently non-labeled biopolymer, poly(SBE-*co*-Az-MAAm), was synthesized in parallel. Aryl azide (Az) moieties were quantified by UV/Vis spectroscopy (Appendix A). We were able to confirm the incorporation of a photolabile moiety into the copolymer at the expected 5 mol%.

### 3.2. Deposition, Stability, and Microscopic Analysis of the Coating

UV-induced photo-deposition was selected as a grafting method since it does not require special functional groups on the adsorbent surface (e.g., primary amines [52]) or additional reagents (e.g., for ATRP grafting [49]), avoiding possible contamination or altering the mechanical properties of the adsorbents. Furthermore, as the coating is induced by UV light, the internal pore surface should not be considerably affected, maintaining adsorption capacity and selectivity at acceptable levels. The same general coating protocol was employed for all types of polyzwitterions and adsorbents, as described in Section 2 and shown in Figure 2.

By a preliminary FTIR study, we demonstrated that under the experimental conditions, photolysis of aryl azide, i.e., the formation of nitrene, occurred (Appendix A). To obtain a fully covalently crosslinked and anchored coat, we exposed dry pre-coated adsorbents to UV light for 20 min [31,32]. Washing of the UV-crosslinked beads was not only performed for the elimination of non-bonded polyzwitterions, but also for evaluating the coating stability. The concentration of polyzwitterions in the supernatant was quantified by UV/Vis spectroscopy (λ_max_ = 468 nm) after every washing step. Ten washing cycles were performed, and polyzwitterions were undetectable by UV/Vis after the 4th to 6th washing steps. To verify the coating stability, coated DALI-pSBE beads were extensively washed with saline solution under static conditions, or under flow conditions with 4% bovine serum albumin in PBS (Appendix A). To assess the effect of UV crosslinking, pre-coated beads without exposure to UV light were used as a control. Fluorescence microscopy confirmed a significant difference between UV crosslinked and non-UV treated beads upon quantification with “Corrected Total Particle Fluorescence” (Appendix A) [53]. Coated beads exposed to UV light appeared significantly brighter, indicating that their coat density was higher than for beads that were washed directly after physical pre-coating. Coating characteristics, specifically homogeneity and thickness, on DALI and CG161c beads were evaluated by confocal laser scanning microscopy [54]. Coatings on the hydrophilic polyacrylate-based DALI beads appeared uniform, whereas brighter spots were occasionally observed on the hydrophobic polystyrene-based CG161c beads (Figure 3 and Appendix A). Nevertheless, in all cases, the beads were completely coated and showed a coating thickness of 5–8 µm.

For further confirmation, thin sections of CG161c beads were examined by fluorescence microscopy. On the samples of coated CG161c beads, clear green rings were revealed, confirming the coating at the outer surface (Figure 4). The thickness of the fluorescent coat was approximately 5 µm, in agreement with data obtained by confocal laser scanning microscopy. Due to autofluorescence, thin sections of white non-transparent DALI beads were not suitable for fluorescence microscopy evaluation.

### 3.3. Effect of Coating on Blood Compatibility

To assess the impact of coated and uncoated polymers on the coagulation parameters, the adsorbents were incubated with fresh citrated human plasma, and the coagulation parameters were determined as described in Section 2 (Figure 5).

Both, uncoated DALI and CG161c resulted in a prolonged PT, indicating a depletion of coagulation factors of the extrinsic pathway. Coating of CG161c, regardless of the nature of the zwitterion applied, significantly reduced PT to levels within the reference range for healthy individuals, whereas coating of DALI had no significant effect on PT. Incubation of plasma with DALI resulted in a substantial prolongation of aPTT, indicating an impact on the intrinsic part of the coagulation cascade. This effect was significantly reduced by coating, but aPTT still remained outside the reference range. None of the CG161c-based adsorbents had a notable effect on the aPTT, which remained in the reference range for all adsorbents, including uncoated CG161c. Analogous to PT, the coating of CG161c resulted in a significantly reduced aPTT. Thrombin time (TT)**,** the time required for clot formation upon the addition of thrombin that converts fibrinogen into fibrin, was elevated upon the treatment of plasma with both, DALI and CG161c, but remained within the reference range. Coating, regardless of the zwitterion applied, significantly reduced TT for all adsorbents, except CG161c-pSBE. Fibrinogen levels were insignificantly reduced for all adsorbents as compared to the untreated control, indicating that the deposition of fibrinogen on the polymers was negligible. AT-III, in contrast, was adsorbed by uncoated DALI and, in particular, by uncoated CG161c, where AT-III levels dropped to 60% of the initial concentration. Coating significantly reduced the depletion of AT-III for both polymers, and AT-III levels remained in the reference range for all DALI-based adsorbents and for CG161c-based adsorbents with polycarboxylic coatings. Protein C was not significantly affected by any of the DALI-based polymers, while it was reduced to 55% of the untreated control by CG161c. Coating of CG161c resulted in significantly diminished protein C adsorption with protein C levels within the reference range. Overall, polyzwitterionic coating of the hydrophobic CG161c polymer beneficially affected coagulation parameters including PT, aPTT, AT-III, and protein C, while this effect was less pronounced for the DALI beads.

To further evaluate the influence of the coating, we performed batch experiments to assess the adhesion of blood cells to CG161c. After the incubation of coated CG161c beads in whole blood, we did not observe significant differences in platelet and red blood cell counts, between unmodified and coated CG161c. However, we detected significant differences in the leukocyte count, mainly attributed to the neutrophil count. For CG161c-pCBMA-C5, the adhesion of neutrophils was suppressed and remained at the level of the control. pSBE or pCBMA-C1 coats did not show such an effect, and no difference in blood cell count was observed in comparison to unmodified CG161c (Figure 6). SEM micrographs of adsorbents employed in this test confirmed these findings. No leukocytes were detected on the surface of CG161c-pCBMA-C5 beads (Appendix A).

In addition to assessing the blood compatibility under static conditions, we studied the interaction of the pSBE-coated DALI adsorbent with whole blood in a dynamic setup. For this study, we selected pSBE-DALI since the pSBE coat showed antifouling behavior toward AT-III (Figure 5E) and antifouling properties of polysulfobetaines are well reported [55,56]. The experimental setup was downscaled from the clinical application for lipoprotein apheresis (Appendix A) [9]. During the recirculation of whole blood over the adsorbent, we monitored blood cell counts every 30 min, assessed platelet activation and the release of extracellular vesicles, as well as the deposition of plasma proteins on the polymer surface (Appendix A). We did not observe significant differences between uncoated DALI and pSBE-DALI in any of these parameters and no differences were observed in SEM images (Appendix A).

The synthesized polymers (1.0 mg/mL) did not induce hemolysis (Appendix A). The same was true for coated adsorbents (10% *v*/*v*). In all cases, cell-free hemoglobin remained below 5 mg/dL, which is considered a physiological level [57,58].

### 3.4. Adsorption of Cytokines and LDL

IL-6 and TNF-α were selected as representative target molecules to evaluate the adsorption capacities of coated CG161c. CytoSorb, which is clinically applied to deplete cytokines, was used as a reference. Adsorption experiments were performed using LPS-stimulated whole blood, as described in Section 2. The efficiency of adsorption is indicated by the percentage of remaining cytokine levels after adsorbent treatment (Figure 7) [45,59].

This approach was chosen to account for donor variations, leading to different cytokine levels after stimulation of whole blood with LPS (IL-6 concentration range: 5.5–24.1 ng/mL, TNF-α concentration range: 13.7–50.0 ng/mL). Uncoated CG161c efficiently depleted IL-6 as well as TNF-α, while the adsorption capacity was reduced by surface modification. To further analyze this reduction, we calculated the relative adsorption capacity (%) of the coated CG161c as
$$\frac{ng\;of\;cytokine\;adsorbed\;per\;ml\;of\;coated\;adsorbent\;}{ng\;of\;cytokine\;adsorbed\;per\;ml\;of\;uncoated\;CG161c\;}\times 100\%$$

Surface coating significantly reduced (Table 1) the adsorption capacity of CG161c to levels comparable to the adsorption obtained with CytoSorb. CG161-pSBE and CG161c-CBMA-C5 reached 67% vs. 63% (IL-6) and 29% vs. 35% (TNF-α) of the adsorption capacity of uncoated CG161c, while coating with pCBMA-C1 reduced the adsorption capacity to 39% for IL-6 and 25% for TNF-α.

To assess the influence of the coating on the adsorption capacity of DALI, whole blood spiked with LDL was incubated with the adsorbents as described in Section 2.10. Uncoated DALI bound 12 ± 0.5 mg LDL per mL of adsorbent, which was reduced to 10 ± 0.1 mg LDL per mL of adsorbent for coated DALI-pSBE beads, i.e., DALI-pSBE maintains approx. 80% of the adsorption capacity of uncoated DALI adsorbent.

## 4. Discussion

Adsorbents for hemapheresis must meet particular requirements beyond their ability to bind and deplete their target molecules, since they are in direct contact with human whole blood, and multiple activation cascades can be triggered on the blood–biomaterial interface. Our coating strategy was thus driven by the goal to achieve a trade-off between enhanced blood compatibility and sufficient adsorption capacity of the coated polymers. We considered that the characteristics of the coating could be influenced by its chemical composition as well as by its deposition on the polymer beads. Regarding the former, we used different zwitterionic coatings, as they resist non-specific interactions with biological systems. The strength of the permanent dipole moment of zwitterions largely depends on the inter-charge distance. All polyzwitterions used in this study bear an identical positively charged dimethyl quaternary ammonium cation. The carboxybetaine monomer CBMA-C1 with a short C1 spacer possesses the structure of an intramolecular salt, whereas the CBMA-C5 monomer with its C5 spacer adopts a real zwitterionic structure with separated ions [60]. When the two charged centers are separated by only a single methylene group, as in betaine CBMA-C1, the low pKa of ≈2.0 indicates that it is difficult to protonate the anionic head group. In the case of betaine CBMA-C5, the pKa will be ≈4.2, which is close to a state of infinite separation of the charged groups in a zwitterionic molecule [28]. In the case of polysulfobetaine, the remaining negative charge is expected as sulfonate (RSO_3_^−^) that is derived from stronger acid than (RCO_2_^−^); furthermore, it is less hydrophilic than carboxylate. We therefore expected different outcomes in terms of coating characteristics and blood compatibility of the coated polymers, e.g., regarding the deposition of plasma proteins on the polymer surface [61]. Recent achievements in the field of polyzwitterions suggest that their specific fouling resistance depends on their chemical structure. The precise dipole orientation of the charged groups has only secondary importance [56].

All polyzwitterions were synthesized by free radical polymerization, and represent statistical terpolymers with zwitterionic (94%), photolabile (5%), and fluorescent (1%) moieties (Figure 1). To minimize potential steric effects influencing copolymerization parameters, we selected the NBD fluorescent label, since the molecular weight of the NBD-MA monomer is comparable to Az-MAAm as well as to the zwitterionic monomers. We refrained from determining the absolute molecular weight of the synthesized polymers since the polymer chains were mutually crosslinked in the following photodeposition step, anyway. Furthermore, there is a lack of zwitterionic standards for size exclusion chromatography, and the use of pullulans as reference materials is questionable [62]. Beyond meeting the balance of enhanced blood compatibility and reduced adsorption capacity or loss of selectivity, any coating technique should be applicable to adsorbents with different chemical compositions, and coating should not affect the bulk properties of the adsorbent. To fulfil these requirements, we employed photodeposition promoted by UV light, yielding thin coatings with a crosslinked network structure covering the external surface of the adsorbent beads. An advantage of this method is the formation of new covalent bonds between substrate and deposited polyzwitterion in contrast to other graft-to methods based on hydrophobic interactions or polydopamine anchors where the long-term stability in the blood is questionable [63]. When evaluating the stability of the coating under static and dynamic conditions in the presence of 4% albumin, we were able to confirm that all beads were covalently coated, and that the crosslinked coat was stable enough to support conditions employed during hemapheresis in clinical settings (Appendix A). The observation of a clearly bordered ring in fluorescent images of thin sections of CG161c adsorbent beads (Figure 4) confirmed that the coating occurred on the bead surface only. DALI beads were not suitable for this evaluation due to their autofluorescence. We anticipated that it would be challenging to achieve homogenous coatings of hydrophobic CG161c beads with hydrophilic polyzwitterions. Indeed, brighter and darker areas were observed in the case of pCBMA-C1. This can be caused by self-aggregation of pCBMA-C1 (brighter spots), the darker spots are either uncoated or the label was quenched due to the configuration of the polyzwitterion in these regions. In the case of pSBE or pCBMA-C5, however, almost uniform coatings were observed in confocal laser scanning and fluorescent microscopy. EDX analysis (SI, Appendix A) revealed a homogenous distribution of carbon, nitrogen, oxygen, and sulfur, supporting the homogeneity of the coatings. For the CG161c-based adsorbents, the presence of nitrogen and the decrease in the percentage of carbon confirmed successful surface functionalization with polyzwitterions. The detection of sulfur in both DALI and CG161c indicated the deposition of a sulfobetaine coat; however, the percentage of sulfur was at the lower limit of detection. In this context, the possibility of partial false-negative results should be considered, as the penetration of the electrons through the soft and porous material could be higher than the thickness of the zwitterionic coat, and thus, a part of the analyzed X-rays may be derived from the uncoated core material. Overall, we were able to confirm that our approach yielded uniform coatings with a thickness of approximately 5 µm for all studied adsorbent-zwitterion combinations.

To assess the blood compatibility of the coated polymers, we focused on coagulation parameters including PT, aPTT, and TT, as well as on the adsorption of AT-III, fibrinogen, and protein C (Figure 5). The prothrombin time (PT) detects alterations of the extrinsic and the final common pathways of the coagulation cascade (Factors II, VI, VII, and X). It was significantly prolonged in the presence of both DALI and CG1616c, pointing to the depletion of the above-mentioned coagulation factors. The use of coated CG161c, regardless of the polyzwitterionic coating applied, resulted in PT values in the reference range, demonstrating reduced protein adsorption on the coated adsorbents. The activated partial thromboplastin time (aPTT) reflects the intrinsic and the common pathways of the coagulation cascade (Factors XII, XI, IX, and VIII). DALI is generally considered highly blood compatible and thus we did not anticipate the pronounced prolongation of the aPTT induced by DALI. However, substantial prolongations of the aPTT have previously been observed in clinical use immediately after DALI treatment [34,64]. This influence of DALI on the intrinsic coagulation pathway can be explained by the activation of Factor XII by contact with the negatively charged surface of the DALI adsorbent. Coating partially masked the negative surface charge and thereby reduced aPTT in the case of pSBE and PCBMA-C5. Fibrinogen adsorption is considered a crucial parameter for the blood compatibility of biomaterials, as it triggers platelet adhesion [65]. Fibrinogen remained in the reference range for all combinations of adsorbents and coatings tested, with a slightly higher decrease for uncoated DALI. AT-III is the plasmatic inhibitor of thrombin as well as Factors IX and X. Its decrease induced by uncoated DALI could be caused by electrostatic interactions between the negatively charged DALI polymer with positively charged residues at the helices A and D as well as the *N*-terminus of AT-III, while coated DALI beads did not result in decreased AT-III levels. Zwitterionic coatings mask the original negative charge of DALI and do not favor AT-III adsorption due to their highly charged, but overall neutral, character. Uncoated CG161c strongly reduced AT-III levels, most likely via hydrophobic interactions. Coating significantly diminished the depletion of AT-III, with polycarboxylic coatings having a slightly stronger effect. Polysulfobetaine coats retain their overall negative charge (RSO_3_^−^), which could be responsible for minor electrostatic interactions with positively charged domains of AT-III. Protein C was strongly depleted by CG161c. Since CG161c is a neutral polymer, its binding of protein C is most likely mediated by hydrophobic interactions. As with AT-III, the coating of CG161c significantly reduced protein C adsorption, and protein C levels remained in the reference range for the coated beads. Wang and co-workers did not observe any significant influence on PT, aPTT, and TT of polycarboxybetaine or polysulfobetaine coats deposited on poly(ethylene terephthalate) (PET) sheets, highlighting the role of the core material [66].

Overall, polyzwitterionic coating enhanced the blood compatibility of CG161c-based adsorbents due to the reduction in protein adsorption. Differences between polysulfobetaine and polycarboxybetaine coatings were observed regarding the effect on TT and AT-III, suggesting that polycarboxybetaines are more appropriate as coating materials. We did not find significant differences for polycarboxybetaine coatings with varied spacer lengths (C1 vs. C5). The difference between polycarboxybetaine coats was detected in batch experiments with whole blood. In the case of CG161c-pCBMA-C5, the adhesion of white blood cells, mainly neutrophils, was completely suppressed and remained at the level of the control (Figure 6). Plain polystyrene can activate leukocytes [67] and induce neutrophil extracellular trap formation [68]. We suppose that pCBMA-C5 effectively suppresses the activation of neutrophils by masking the polystyrene CG161c core.

Next to the beneficial effect of CG161c coating in terms of enhanced blood compatibility, our data confirm that coating is associated with a reduced adsorption capacity. Uncoated CG161c depleted cytokines with high efficiency but, due to its pronounced hydrophobicity, is not suited for clinical use in hemosorption [44,69,70]. Like CG161c, CytoSorb, which was used as a reference in our study, is a polystyrene-based polymer, coated with polyvinylpyrrolidone to enable its clinical application. While CytoSorb exhibited significantly lower adsorption of IL-6 and TNF-α than uncoated CG161c, there was no significant difference in CytoSorb vs. CG161c-pSBE (Figure 7). CG161c-pCBMA-C5 exhibited higher adsorption capacity toward TNF-α than CytoSorb.

The effect of polyzwitterionic coating was less clear for DALI-based adsorbents, except for the moderate influence on TT, aPTT, and AT-III. To evaluate the effect of the coating in more detail, we studied the interaction of polysulfobetaine-coated DALI with whole blood by monitoring the blood cell count and release of extracellular vesicles (EVs) in a dynamic model (Appendix A), but no significant impact of the coating on the blood cell count or release of EVs was detected (Appendix A). Since DALI is highly blood compatible per se, the potential of further improving its blood compatibility by polyzwitterionic coating is limited.

## 5. Conclusions

Our study suggests that polyzwitterionic coating significantly improves the blood compatibility of hydrophobic adsorbents. Based on our findings, polycarboxybetaine pCBMA-C5 is the most promising polyzwitterion for the coating to be used in whole blood. Its C5 spacer ensures the separation of charged groups in a zwitterionic molecule, resulting in balanced physical properties such as hydrophilicity, pKa, capacity to form homogenous coats and blood compatibility evident by a reduction in coagulation times (PT, aPTT, TT), lowered protein adsorption (AT-III, protein C), and suppressed adhesion of neutrophils. Adsorption capacities of CG161c-pCBMA-C5 toward cytokines (IL-6, TNF-α) were comparable to CytoSorb. The nature of the coating is critical to achieving a trade-off between adsorption capacity and blood compatibility.

## Data Availability

Not applicable.

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
