# Peer review of "Polyzwitterionic Coating of Porous Adsorbents for Therapeutic Apheresis"

_jfb, 2022, doi:10.3390/jfb13040216_

Round 1
Reviewer 1 Report
In this study, Semak et al. reported a blood compatible matrices via surface modification with polyzwitterionic polysulfobetainic and poly carboxybetainic (pCBMA-C5) coatings that significantly improved the blood compatibility of hydrophobic adsorbents. Their results suggest that pCBMA-C5was the most promising polyzwitterion for the coating. The following issues can be resolved before publication:
1. The author should describe certain numerical results in the abstract, e.g., absorption capacity for the cytokines and blood compatibility, etc of the proposed coating compared to control groups. About the advantages of this product.
2. How does the inner coating of NBD monitor the deposition and stability of the coating on the bead surface?
3. What is the function of photolabile moiety?
4. “both, DALI and CG161c, but remained within the reference range for both adsorbents.” What is the reference range?
5. The discussion seems quite long and confusing. It can be shortened as some of the points have already been mentioned in the result sessions.
Author Response
Dear Editor Prof. Stefano Bellucci,
Dear Reviewers and MDPI/JFB Team,
Thank you very much for considering our manuscript (jfb-1995511) titled “Polyzwitterionic Coating of Porous Adsorbents for Therapeutic Apheresis”. We sincerely thank the editor and all reviewers for their valuable feedback. According to the reviewers’ comments, we have made modifications to our manuscript. The manuscript has been greatly improved as a result of the editor and reviewers’ efforts. Our point-by-point actions and replies to the referees’ comments are attached as separate attached pdf documents.
We would be very grateful if the revised manuscript could be considered for a publication in Journal of Functional Biomaterials.
Sincerely yours,
Vladislav SEMAK (corresponding author)
E-mail: vladislav.semak@donau-uni.ac.at

Reviewer 2 Report
Comments and Suggestions for Authors
This work developed blood compatible matrices by surface modification with polyzwitterionic polysulfobetainic and polycarboxybetainic coatings, for the porous adsorbents for therapeutic apheresis. The ideal is novel and the work is interesting. I think this paper can be accepted and some mirror suggestion is shown below.
1 The performance comparison of this novel porous adsorbents with other reported adsorbents can be tabled and discussed.
2 The adsorbents can be reused or not? Maybe the regeneration or treatment of the used adsorbents could be mentioned in the paper.
Author Response

(The authors gave the same response as above.)

Reviewer 3 Report
This manuscript does the symmetrical study of blood compatibility of commercial adsorbents coated with two series of polyzwitterionics. It is very interesting with good results and writing. It can be accepted with revision based on the following suggestions.
1) In this study, “hydrophobic polystyrene-co-divinylbenzene and hydrophilic polyacrylamide-co-polyacrylate beads” in study is commercial adsorbent of CG161c and CytoSorb, respectively. If being impossible in manuscript title, this information should be mentioned in Abstract.
2) Line 149, 168 and so on: as for “adsorbents (200 μL per tube)”, it should be bead suspension. If so, offer the weight concentration. In subsection 2.10, the amount of used absorbent should be mentioned. In Line 408, the concentration of bead adsorbent should be also mentioned.
3) Subsection 3.1: GPC results for molecular weights of different synthesized polymers are encouraged to offer.
4) Line 261 and 262: even if azido group is used for grafting reaction, there is still certain reactive groups in the substrate, among which amine lies.
5) Line 278: the statement that “polyzwitterions were undetectable” should be specified.
6) Figure 4: what are the upper photos?
7) Line 415 and 416: adsorption of blood cell and protein onto one adsorbent suggests the bad blood compatibility of this adsorbent. Thus, “trade-off” is difficulty to follow.
8) Line 431: as for “it is less hydrophilic than carboxylate”, it suggests that sulfonate (RSO3-) anion is less hydrophilic than carboxylate anion, right? But, this suggestion is wrong.
9) Line 435-436: “dipole orientation” is in the range of chemical structure, thus “rather than” is improper.
10) Line 439: what kind of “copolymerization parameters” have what kind of “potential effects”? Please specify both of them.
11) Line 440: pay attention to that nitro-aromatic compound is well-known as the inhibitor of free radical polymerization.
12) Line 442: as for “purified by dialysis against water”, how about the solubility of different monomers in water?
13) Line 443-446: before UV crosslinkage, it is absolutely possible to measure the relative molecular weight of polymers with PEG as standard by GPC.
14) The paragraph across Page 13 and 14: as for “the observation of a clearly bordered ring in fluorescent images”, “due to the strong material autofluorescence” and so on, the related figures are suggested mentioned.
15) Line 514: as for “CG161c is a neutral polymer”, what does it mean? CG161 is polyzwitterionic.
16) “polyacrylic acid” (L105) → “poly(acrylic acid)”; “polyethylene terephthalate” (L519) → “poly(ethylene terephthalate)”.
Author Response

(The authors gave the same response as above.)

Reviewer 4 Report
The paper entitled "Polyzwitterionic coatings of porous adsorbent or therapeutic apheresis" contains valuable information about the preparation of blood-compatible coated adsorbents. The experimental plan is well-defined, the evaluations are correct. Changing the spacer in the main monomer units showed the importance of steric parameters. The products testing consists of well-organized experiments, and the discussion is also clear and contains valuable results.
I think the work is interesting, the work is well-organized and the mansucript is well-written, thus I can suggest accepting to publish it in IJMS.
Some minor remarks:
Abbreviations "DALI" and CG161c (Figure 1) should be given at the place of first using (Fig.1), not only in chapter 2.1
line 222, instead of “antisymmetric carboxylate [(COO)- st as]”, I suggest giving "antisymmetric C=O stretching mode (nas) of carboxylate ion".
The structure of the polymer given does not show the statistical distribution of each component in it. It looks like a polymer with 94 zwitterionic monomers, five azides, and one nitro-group containing monomer, in this order, repeating these units in this order.
I suggest giving a general formula when giving only one monomer unit but giving a -C(=O)-R group -instead of the C(=O)-X, C(=O)-N and C(=O)-O groups, and it could be written that R = X-...., N-.. and O-.. groups, in 94:5:1 statistical distribution. Similarly, in the ESI (in the synthesis equations).
The common symbols for symmetric/antisymmetric stretching/deformation modes should be used in the ESI as well.
Author Response

(The authors gave the same response as above.)
